

# Evaluation of the forces applied by rubber dam clamps on mandibular first molar teeth with different endodontic access cavities: a 3D FEA study

Mehmet Eskibağlar[1], Serkan Erdem[2], Büşra Karaağaç Eskibağlar[3] and Mete Onur Kaman[2]

[1] Department of Endodontics, Faculty of Dentistry, Firat (Euphrates) University, Elazığ, Turkey
[2] Department of Mechanical Engineering, Engineering Faculty, Firat (Euphrates) University, Elazığ, Turkey
[3] Department of Pediatric Dentistry, Faculty of Dentistry, Firat (Euphrates) University, Elazığ, Turkey

## ABSTRACT

**Background:** This study aimed to examine the effect of the force applied by rubber dam clamps made from different materials on mandibular first molar teeth with various designs of endodontic access cavities using finite element analysis.

**Methods:** A intact tooth (IT) and seven different endodontic access cavities namely, a traditional endodontic cavity (TRADAC), a guided endodontic cavity (GEC), a conservative endodontic cavity (CAC), an ultra-conservative access cavity (UAC), a truss access endodontic cavity (TRSAC), a mesial caries access cavity (MCAC), and a distal caries access cavity (DCAC), along with two different clamp finite element models, were created. The clamp models were made of polyether ether ketone (PEEK) and stainless steel (SS). The forces applied by the clamps were calculated based on the axial section distance of the tooth, and these forces were applied to the contact areas on the tooth. Stress distribution models were calculated using maximum von Mises (vM) stress.

**Results:** The lowest vM stress under the forces applied by the SS and PEEK clamps was found in the IT model (80.914 MPa) with the PEEK clamp. The highest vM stress was found in the DCAC model (759.49 MPa) applied with the SS clamp. The forces applied by SS clamps resulted in higher vM stress values in every cavity design than those applied by PEEK clamps.

**Conclusion:** PEEK clamps generated less force than SS clamps. However, clinicians should follow various isolation strategies (clamp made of different materials, split dam, *etc.*) according to different cavity types of the tooth.

Corresponding author
Mehmet Eskibağlar,
meskibaglar@firat.edu.tr

## INTRODUCTION

Endodontic treatment is one of the most effective options for the management of pulp and periapical diseases. Such treatment often involves the preparation, cleaning, and filling of the root canal system (*Jiang et al., 2018*). The initial step of endodontic treatment is the

preparation of the access cavity (*Silva et al., 2020*), which is crucial for the effective instrumentation and obturation of root canal therapy and also plays a key role in the healing of pulpal and periapical infections (*Jiang et al., 2018*; *Silva et al., 2020*).

In recent years, the emergence of minimally invasive endodontic (MIE) access cavity designs has advocated preserving as much healthy dentin as possible (*Wang et al., 2023*). These designs vary in complexity from those that involve removing the entire roof of the pulp chamber to more challenging orifice-directed approaches (*Silva et al., 2022*). Various designs for MIE cavities have been proposed, including the conservative access cavity (CAC), the ultra-conservative access cavity (UAC), and the truss access cavity (TRSAC), as well as the computer-aided design-guided endodontic cavity (GEC) (*Elkholy et al., 2021*; *Silva et al., 2020*; *Wang et al., 2023*).

One of the critical factors for the success of endodontic treatment is isolation, with the recommended method for this purpose being a rubber dam. The clamps used in the application of a rubber dam secure fixation to the tooth (*Ahmad, 2009*; *Bhuva, San Chong & Patel, 2008*). Clamps are typically made of stainless steel (SS) materials (*Bhuva, San Chong & Patel, 2008*), although clamps made of polyether ether ketone (PEEK) (SoftClamp; KerrHaweTM, Bioggio, Switzerland) are also available. For stabilization of the rubber dam, it is desired that the clamps contact the tooth at four points from the cervical region. These contact areas cause stress formation in the pericervical region of the tooth (*Bhuva, San Chong & Patel, 2008*).

Finite element analysis (FEA) is a form of numerical analysis widely used in engineering and has since been adopted for use in dentistry, particularly for modelling bone, teeth, dental restorations (*Choi, Conway & Ben-Nissan, 2014*). FEA is a repeatable method of analysis performed *in silico*. In this analysis, the parameters being investigated can be varied as necessary, even to extremes for the research at hand, without any risk as they are in a simulation environment (*Trivedi, 2014*). FEA analyses stress and deformations and solves complex structural problems (*Chien, Walsh & Peters, 2021*). Numerous studies have been conducted using laboratory and FEA on the structural integrity of teeth with different access cavities in endodontic treatments (*Jiang et al., 2018*; *Neelakantan et al., 2018*; *Santosh, Ballal & Natanasabapathy, 2021*; *Wang et al., 2023*). However, to the authors' knowledge, no studies have evaluated the forces exerted by rubber dam clamps on teeth with different access cavities. FEA allows for good control of variables and can isolate some disadvantages of *in vivo* studies, such as sample selection bias, thereby enabling testing of the variables' effect under conditions as close to clinical as possible (*Silva et al., 2021*). This study aims to examine the effect of the force applied by rubber dam clamps made of different materials on mandibular first molar teeth with various endodontic access cavity designs using FEA. Our study's null hypothesis is that the forces caused by clamps made of different materials differ when applied to different cavities.

## MATERIALS AND METHODS

Ethical approval was obtained from the Firat University Non-Interventional Research Ethics Committee (session number 2023/09-36) Written consent was obtained from

eligible patient whose mandibular first molar was planned to be extracted due to advanced periodontal disease and orthodontic treatment.

## FEA model generation

A caries-free, mature mandibular first molar tooth was scanned using a SkyScan 1272 high-resolution micro-CT scanner (SkyScan, Aartselaar, Belgium) with a 14 μm voxel size. The data were combined using CTAn software (SkyScan), and a three-dimensional (3D) model was created with a *.stl extension. Models with the *.stl extension were edited in Geomagic Design software (Geomagic, Inc., Research Triangle Park, NC, USA) through reverse engineering and saved as *.prt files. The tooth model with a *.prt extension was then opened in Solidworks software (Dassault Systems SA, Concord, MA, USA) and separated into enamel, dentin, and pulp tissues corresponding to its original composition. Then, using the same software, a uniform 0.3 mm layer of periodontal ligament was created around the root (1.5 mm below the cementoenamel junction), followed by modeling the surrounding cancellous and cortical bone in the root.

## Design of the clamps

The dimensions of the SoftClamp posterior (KerrHawe™) and Hu-Friedy 14A clamps (Hu-Friedy, Chicago, IL, USA) were measured using a digital caliper. These clamps were modeled using Solidworks software according to these dimensions.

## Creation of endodontic access cavities in teeth

After generating the model of an intact mandibular first molar tooth, seven different models of endodontic access cavity designs were created in Solidworks software using the extrude cut command, in accordance with the designs described in the literature (*Silva et al., 2020*) (Figs. 1A–1H).

(B) The mesial caries access cavity (MCAC) model: An endodontic access cavity was opened to reach the pulp chamber of the tooth, assuming the presence of caries or a restoration on the mesial surface of the tooth.

(C) The distal caries access cavity (DCAC) model: An endodontic access cavity was opened to access the pulp chamber of the tooth, assuming the presence of caries or a restoration on the distal surface of the tooth.

(D) The traditional access cavity (TRADAC) model: The roof of the pulp chamber was completely removed, and an endodontic access cavity was opened to provide direct access to the canal orifices.

(E) The conservative access cavity (CAC) model: A central endodontic access cavity was opened on the occlusal surface of the pulp to access the root canal orifices.

(F) The truss access cavity (TRSAC) model: Two separate endodontic access cavities were opened to access the root canal orifices.

(G) The ultra-conservative access cavity (UAC) model: An endodontic access cavity was opened while preserving as much of the roof of the pulp chamber as possible.

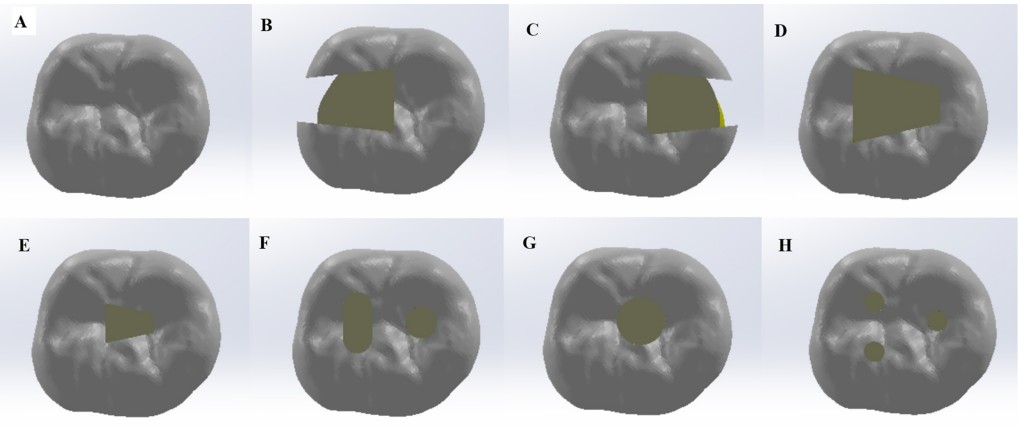

**Figure 1 Representative schematic diagrams of different unite element models.** (A) IT (intact tooth), (B) MCAC (mesial caries access cavity), (C) DCAC (distal caries access cavity), (D) TRADAC (traditional access cavity), (E) CAC (conservative access cavity), (F) TRSAC (truss access cavity), (G) UAC (ultra-conservative access cavity) and (H) GEC (guided endodontic cavity) models.

(H) The guided access cavity (GEC): Three separate endodontic access cavities of minimal size were drilled for straight-line access to the root canal orifices.

For the endodontic access cavities, the volume of material removed was 119.519 mm³ for MCAC model, 131.087 mm³ for DCAC model, 131.887 mm³ for TRADAC model, 35.719 mm³ for CAC model, 66.136 mm³ for TRSAC model, 45.152 mm³ for UAC model, and 28.806 mm³ for GEC model.

## Meshing and set material properties

The models with different endodontic access cavities and clamps were subjected to meshing using ANSYS Workbench software. Following this process, the number of tetrahedral elements and nodes for the models are provided in Table 1.

The mechanical properties of the tooth, periodontal tissue, and clamps used in the FEA are shown in Table 2. The models used in the FEA were assumed to have homogeneous, linear, and isotropic properties. Additionally, perfect bonding between each component was assumed.

## Calculation of the forces applied and strain measurement

The clamps were positioned approximately 1 mm below the equatorial line of the mandibular first molar tooth. The areas where the clamps would contact the teeth were determined to be approximately 0.34 mm² on the mesial surface and 0.76 mm² on the distal surface of the tooth both by marking on the scanned tooth model and through overlay processing in Solidworks software. The distance required for the clamps to be opened at the specified distance from the tooth was measured by taking an axial section at the determined distance. It was determined that the opening distance required was approximately 4.51 mm in total on the distal part of the tooth and 5.12 mm on the mesial part ($\Delta CB = 4.51$ mm, $\Delta ED = 5.12$ mm). The clamp models were transferred to ANSYS

**Table 1 Number of elements and nodes in the models.**

|  | IT | MCAC | DCAC | TRADAC | CAC | TRSAC | UAC | GEC |
|---|---|---|---|---|---|---|---|---|
| Elements | 423.295 | 409.518 | 454.144 | 434.175 | 421.403 | 470.634 | 420.377 | 427.593 |
| Node | 246.381 | 237.139 | 269.916 | 254.607 | 244.810 | 279.920 | 244.193 | 248.396 |

**Table 2 Mechanical properties of the investigated materials.**

|  | Young's module (MPa) | Poisson rate |
|---|---|---|
| Enamel (*Sorrentino et al., 2007*) | 84,100 | 0.3 |
| Dentin (*Sorrentino et al., 2007*) | 18,600 | 0.31 |
| Pulp (*Aslan et al., 2021*) | 2 | 0.45 |
| PDL (*Poiate et al., 2011*) | 68.9 | 0.45 |
| Alveolar bone (*Li et al., 2006*) | 13,700 | 0.3 |
| PEEK (*Ouldyerou et al., 2022*) | 3,600 | 0.37 |
| SS | 193,000 | 0.3 |

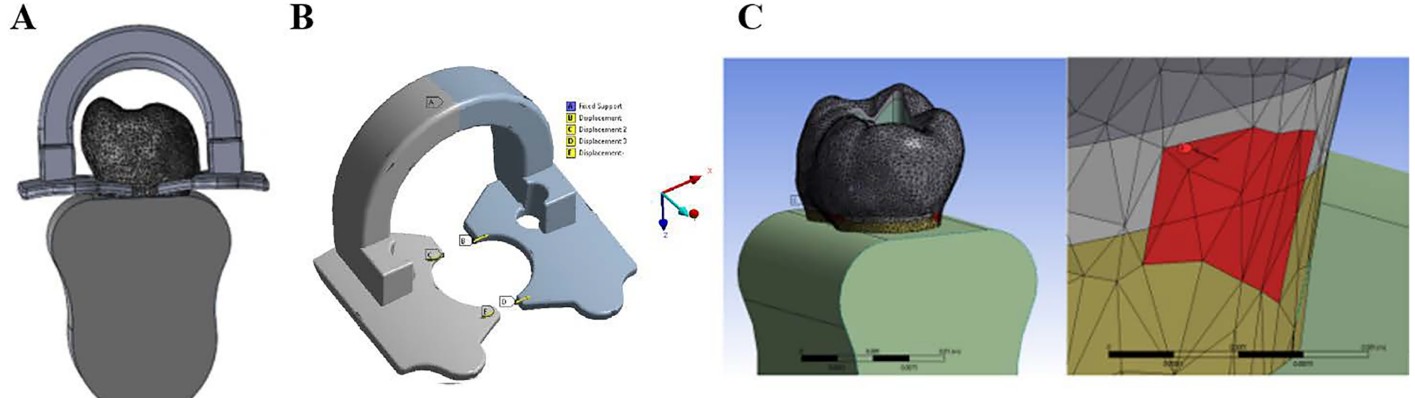

**Figure 2** (A) Determination of the contact area of the clamp and the tooth, (B) Calculation of the force that the clamp will apply to the tooth, (C) Application of the force to the contact areas.

Workbench software, and fixed support was applied to section A of the clamps, while a displacement of ΔCB/2 = 2.255 mm was applied to each of the contact surfaces B and C and that of ΔED/2 = 2.56 mm was applied to each of the contact surfaces D and E. The force required to be applied by the clamps under the specified boundary conditions to the tooth was obtained as a result of FEA. This force was determined to be 58.25 N for the distal surface of the PEEK clamp and 3.75 N for the mesial ends; it was 108.68 N for the distal surface of the SS clamp and 31.28 N for the mesial ends. These forces were applied to the enamel surfaces of the tooth according to the determined contact areas Fig. 2.

To validate the force estimates obtained *via* FEA, a strain gauge (BFLA-5-5; Tokyo Sokki Kenkyujo Co., Ltd., Tokyo, Japan) was used in three directions to calculate the experimental stresses. The clamps were placed on the tooth to determine the contact areas

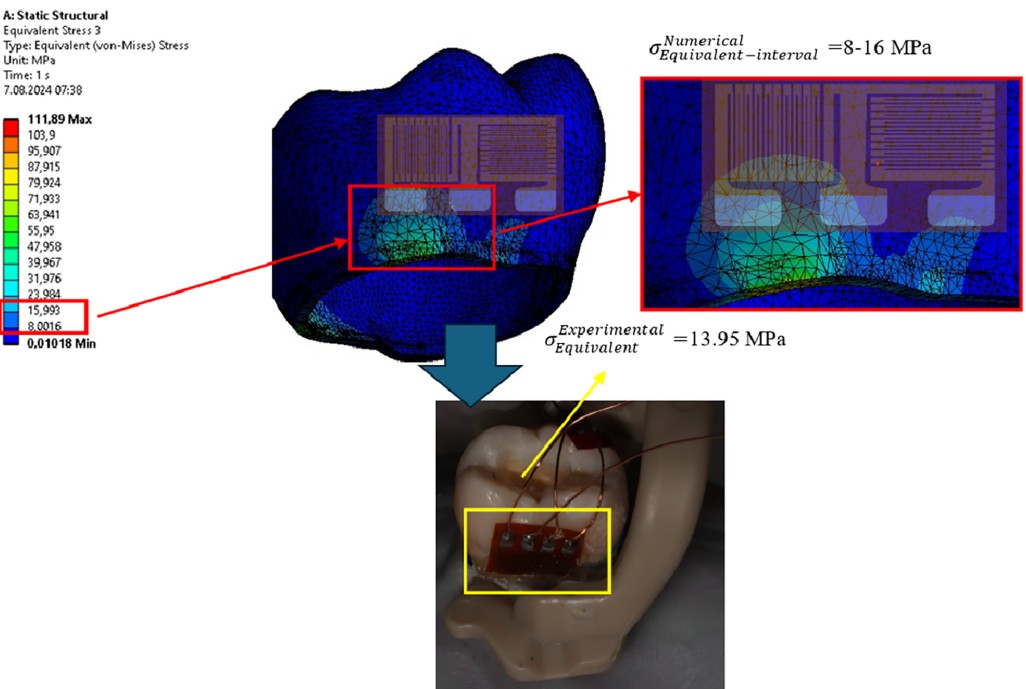

**Figure 3 Demonstration of comparison of FEA with strain gauge experimental setup.**

and the strain gauge was bonded to the enamel of the tooth with an adhesive system (Scotchbond Multi-Purpose, 3M ESPE) so that the clamps did not touch the contact areas. The clamp used in FEA was placed on a mandibular first molar with similar dimensions. The strain gauges were connected to a data acquisition device (VTA-1704; Marmatek Measurement Technologies, Turkey). The tooth clamps were placed so that they did not touch the strain gauge (Fig. 3). The data obtained were recorded on a computer for signal conversion and data analysis.

## RESULTS

When evaluating the forces applied by the clamps, it was observed that von Mises (vM) stress occurred mostly in the enamel layer of the tooth, regardless of the cavity design and type of clamp. The lowest levels of vM stress on the tooth created by the PEEK clamp were found in the IT model (80.914 MPa). According to the cavity designs, the vM stress levels stress created by the PEEK clamp on the enamel layer of the tooth were lowest in the GEC model (108.88 MPa), followed by the CAC (111.89 MPa), UAC (116.15 MPa), TRSAC (130.25 MPa), TRADAC (149.3 MPa), MCAC (215.93 MPa), and DCAC (241.84 MPa) models, in ascending order. The maximum vM stresses that occur in the models are shown in Fig. 4 looking apically at the enamel layer of the tooth. The maximum vM stress levels in the tooth with PEEK clamps forces applied were determined as dentin (2.62 MPa), pulp (0.0038 MPa), periodontal ligament (PDL) (0.0992) and alveolar bone (6.4 MPa). In the experimental setup where PEEK clamp was applied, strain values were similar to those in

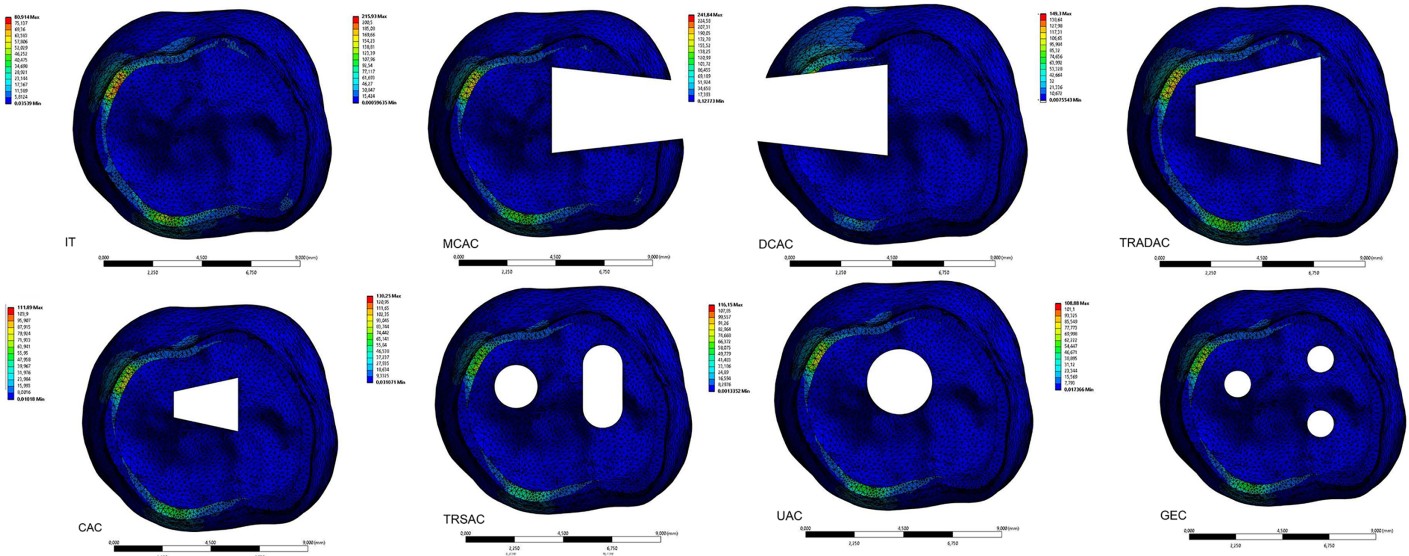

**Figure 4 The effect of the PEEK clamp on the apical view of the maximum vM stress in the enamel tissue of the tooth caused by the force applied to the tooth by the PEEK clamp for different access designs.**

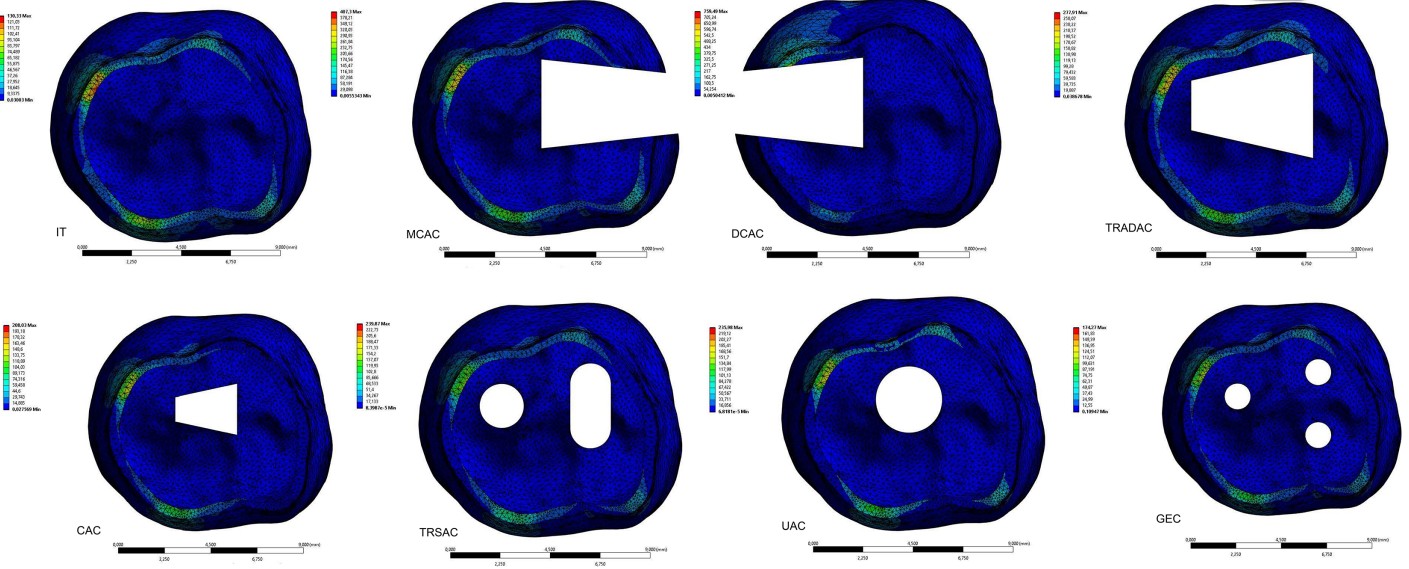

**Figure 5 The effect of the SS clamp on the apical view of the maximum vM stress in the enamel tissue of the tooth caused by the force applied to the tooth by the SS clamp for different access designs.**

FEA: IT (10.98 MPa), MCAC (29.73 MPa), DCAC (32.84 MPa), TRADAC (19.95 MPa), CAC (13.68 MPa) (Fig. 3), TRSAC (17.24 MPa), UAC (15.32 MPa), GEC (13.14 MPa).

Similar to the PEEK clamp, the levels of vM stress created by the SS clamp on the tooth were found to be lowest in the IT model (130.33 MPa). According to the cavity designs, the levels of vM stress created by the SS clamp on the enamel layer of the tooth were lowest in the GEC model (174.27 MPa), followed by the CAC (208.03 MPa), UAC (235.98 MPa), TRSAC (239.87 MPa), TRADAC (277.91 MPa), MCAC (407.3 MPa), and DCAC (759.49

**Table 3 The maximum vM stress level (MPa) in dentin, pulp, pdl and alveolar bone in all cavity types.**

| | Clamp types | |
| --- | --- | --- |
| | PEEK | SS |
| Dentine | 2.62 | 2.62 |
| Pulp | 0.0038 | 0.0064 |
| PDL | 0.0992 | 0.0118 |
| Alveolar bone | 6.4 | 10.32 |

MPa) models, in ascending order. The maximum vM stresses that occur in the models are shown in Fig. 5 looking apically at the enamel layer of the tooth. The maximum vM stress levels in the tooth with PEEK clamps forces applied were determined as dentin (2.62 MPa), pulp (0.0064 MPa), periodontal ligament (PDL) (0.0118 MPa) and alveolar bone (10.32 MPa). In addition, the maximum vM stress levels on the dentin, pulp, periodontal ligament (PDL), and alveolar bone of the tooth are provided according to the clamp types in Table 3. In the experimental setup where the SS clamp was applied, strain values IT (18.74 MPa), MCAC (54.02 MPa), DCAC (104.4 MPa), TRADAC (35.16 MPa), CAC (27.92 MPa), TRSAC (30.58 MPa), UAC (30.66 MPa), GEC (21.88 MPa) were similar to those in FEA.

## DISCUSSION

FEA is an appropriate method for conducting clear and objective testing of biological systems (Prati et al., 2021; Wang et al., 2023). This method serves as a useful tool for examining complex systems and is widely utilized in endodontic stress analyses (Jiang et al., 2018). This study aimed to evaluate the stress applied by clamps to a mandibular first molar tooth with seven different designs of endodontic access cavities, as specified in the literature. We used the FEA method for our analyses to allow for standardization in analyzing different cavity models within the same tooth model (Askerbeyli Örs et al., 2019; Jiang et al., 2018; Saber et al., 2020). Thus, our study evaluated the effects of both rubber dam clamps and endodontic access cavities. The vM stress distribution was used to examine stress distribution and compare our study with other studies.

Various studies have reported that mandibular first molars are the teeth most frequently requiring endodontic treatment. It has also been stated that they break more often than other teeth, which is why our study utilized this tooth (Touré et al., 2011; Zadik et al., 2008). Properly prepared endodontic access cavities positively influence the success of treatment (Plotino et al., 2017). Therefore, our study was planned to include seven different cavity designs, aiming for consistency with existing research (Askerbeyli Örs et al., 2019; Silva et al., 2020). Various studies have evaluated the mechanics properties of different endodontic access cavity designs, using both experimental approaches and FEA. While some of these studies have found statistically significant differences, others have not (Özyürek et al., 2018; Sabeti et al., 2018; Santosh, Ballal & Natanasabapathy, 2021). However, none have assessed the value and distribution of vM stresses created by rubber

dam clamps on different cavity designs. As the application points of the clamps were mostly on the enamel tissue of the tooth, it was found that the maximum vM stress values obtained in all models were mainly on the enamel layer of the tooth, while the vM stress values in the other tissues of the tooth were negligible. This result may be due to the fact that the enamel tissue has a higher Young's modulus than the other tissues of the tooth. This finding is consistent with our previous research using IT to assess the impact of clamps (*Eskibağlar, Erdem & Kaman, 2023*). Additionally, the lowest vM stress from both types of clamps was found on the IT model, while the lowest stress by design was found on the GEC model. Furthermore, the order of increasing vM stress values for the designs was the GEC model, followed by the CAC, UAC, TRSAC, TRADAC, MRAC, and DRAC models. The stress values between the CAC and UAC models, as well as between the TRSAC and TRADAC models, were found to be similar, while higher stress values were observed in the MRAC and DRAC models. Based on the results of our study, the null hypothesis is accepted. The reason why the GEC model has lower stress levels compared to other cavity designs may be due to its design, which provides direct access to the canals and removes the least amount of tooth tissue. The higher stress levels in the MRAC and DRAC models, which do not have marginal ridges, may be due to the lack of marginal ridges. The results of our study are consistent with similar studies (*Nawar et al., 2023*; *Wang et al., 2023*).

Strain gauge is a well-established, non-destructive method with good reproducibility for real-time *in situ* strain measurements (*Tonelli et al., 2021*). To measure strain in the mandibular first molar as a result of clamp application, the strain gauge was attached to the lingual surface of the crown of the tooth. In this way, deformation was measured *in vivo*, as opposed to stress, which cannot be measured directly. The deformation values measured by the strain gauge placed on the lingual surface of the mandibular molar were compared with the values calculated by FEA and showed similar behaviour, thus confirming the FEA result (*de Paula Rodrigues et al., 2017*).

In the isolation of teeth, rubber dam clamps are placed in the pericervical regions of the teeth (*Bhuva, San Chong & Patel, 2008*). Studies evaluating the mechanical properties effects of different forces have associated the fracture resistance of treated teeth with the preservation of the tooth structure (*Eskibağlar et al., 2021*; *Wang et al., 2023*). In particular, it has been noted that the pericervical dentin region of the tooth plays a significant role in the tooth's fracture resistance (*Eskibağlar et al., 2021*; *Saber et al., 2020*; *Wang et al., 2023*). Additionally, biomechanical studies using FEA that evaluated different forces reported that vM stress values were higher in the pericervical dentin region of the teeth compared to other areas (*Jiang et al., 2018*; *Saber et al., 2020*; *Wang et al., 2023*). In our study, the stresses after the application of the clamps occurred in the precervical region of the tooth. SS clamps produced more vM stress than PEEK clamps.

When the clamps used in our study were examined in terms of design, the contact areas on the mesial and distal surfaces of the teeth were produced asymmetrically. This asymmetrical structure and the anatomy of the molar tooth caused variability in the magnitude of force applied by the clamps to their contact areas. Moreover, the modulus of elasticity of the SS material was 50 times greater than that of PEEK. The design differences

in the clamps (*e.g.*, thickness and width) prevented the force from increasing linearly. For these reasons, when the clamps were evaluated, the SS clamps applied more force to the tooth than the PEEK clamps. Studies have reported that the force applied by SS clamps has the potential to cause pain and possible bone necrosis (*Gozon, Rumford & Cervenka, 2023*; *Yoon & Chussid, 2009*). We believe it is important that the clamps contact the enamel of the tooth to avoid possible periradicular damage.

In our study, we evaluated the effects of two similarly sized clamps made of two different materials on mandibular first molars with different cavities. The FEA method we used in this study has disadvantages such as its inability to simulate clinical conditions and the inability to reflect anisotropic structures such as dentin (*Elkholy et al., 2021*; *Saber et al., 2020*; *Wang et al., 2023*). However, manufacturers produce clamps with many different designs for the isolation of teeth. The morphological differences in teeth and the differences in clamp designs can create different stress levels. Additionally, these clamps are sterilized and exposed to various fluids (*e.g.*, sodium hypochlorite, EDTA, oral fluids, or sterilization fluids), and it is considered that these factors could cause deformations in the clamp that could break it (*Sutton & Saunders, 1996*) and alter the force applied to the tooth. These factors are among the limitations of our study.

Based on the results of our study, we recommend to clinicians that, for the isolation of teeth with significant tissue loss, such as restorative cavities, if the adjacent posterior tooth is healthy, the clamp should be placed on that tooth for isolation with a split dam technique. If this is not possible, then a clamp made of PEEK should be applied after the restoration of the tooth to be isolated.

## CONCLUSIONS

Within the limitations of this study, PEEK clamps generated lower vM stress values in all cavity designs than SS clamps. The maximum vM stress value in the tooth varied in terms of cavity designs. The vM stress levels generated occurred mostly on the enamel layer of the tooth. The amount of tooth tissue removed influenced the magnitude of vM stress.
It should be borne in mind that the force applied to the tooth by the clamps is for a limited time and that these forces are not occlusal. However, we believe that more studies should be carried out on this subject.

### Funding
The authors received no funding for this work.

### Competing Interests
The authors declare that they have no competing interests.

### Author Contributions
- Mehmet Eskibağlar conceived and designed the experiments, analyzed the data, authored or reviewed drafts of the article, and approved the final draft.

- Serkan Erdem conceived and designed the experiments, performed the experiments, analyzed the data, authored or reviewed drafts of the article, and approved the final draft.
- Büşra Karaağaç Eskibağlar performed the experiments, prepared figures and/or tables, authored or reviewed drafts of the article, and approved the final draft.
- Mete Onur Kaman performed the experiments, analyzed the data, authored or reviewed drafts of the article, and approved the final draft.

### Human Ethics

The following information was supplied relating to ethical approvals (*i.e.*, approving body and any reference numbers):

Ethical approval was obtained from the Firat University Non-Interventional Research Ethics Committee (session number 2023/09-36).

### Data Availability

3D models of the images in our article are available at Cults:

https://cults3d.com/en/3d-model/various/ss-clamp-model-meskibaglar

https://cults3d.com/en/3d-model/various/peek-clamp-model-meskibaglar

https://cults3d.com/en/3d-model/various/tooth-model-meskibaglar

https://cults3d.com/en/3d-model/various/jaw-and-tooth-model-meskibaglar

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
