# Peer review of "Evaluation of the forces applied by rubber dam clamps on mandibular first molar teeth with different endodontic access cavities: a 3D FEA study"

_PeerJ, doi:10.7717/peerj.17921_

## Round 0.1 · original submission · Major Revisions

Dear Authors,

Thank you for submitting your manuscript "Evaluation of the Forces Applied by Rubber Dam Clamps on Mandibular First Molar Teeth with Different Endodontic Access Cavities through Finite Element Analysis." Major revisions are required. The principal topics are: shorten the title, introduce a null hypothesis and justify the use of FEA in the introduction. Improve Figures 1, 3, and 4 for clarity and ensure consistent perspectives. Explain the validation of the FEA models and address the limitations of using linear, homogeneous, and isotropic materials in the methodology. Explain about using non-linear contact between clamps and enamel. Organize the discussion better and remove strong clinical claims about SS clamps and tooth fractures from the conclusion.

Please revise your text accordingly and resubmit.
Kind regards,
Dr. Tribst JPM

Reviewer 1 ·

Basic reporting

Concerning the abstract:
Please add the acronym you’ll use for the Traditional access cavity in the methods section like you did with the others.

Concerning the introduction:
Generally, the introduction is concise and to the point.
- “..no study has evaluated the forces applied by rubber dam clamps to teeth..” I praise the authors for seeking a knowledge gap, though I’d advise the use of more conservative wording such as (To the knowledge of the authors, there are no studies that evaluate etc).

Concerning the methodology:
Figure 1 can be much better. Screenshots from the actual models with the restoration removed would be more aesthetic. Also, the angles of the cavities are very sharp. This wouldn’t probably affect your results, but only because loading isn’t occlusal in your scenario. If the study simulated chewing these sharp-cut angles would cause bizarre results.

Concerning results:
- Generally, the results need better presentation. Figures 3 and 4, should probably be amalgamated together having corresponding models above each other to facilitate comparisons.
- Also, the perspective of figures 3 and 4 is not easy to recognize at all. I mean even the figure legend doesn’t explain the view from which the screenshots were taken as being the enamel shown from an apical view (If I’m wrong about that, then this really needs clarification). So maybe we can have 2 more figures showing the aspects on which the clamps rest?
- Also, in the same figures 3 & 4, not all models are shown from the same perspective.

Concerning the discussion:
The discussion isn’t as well-organized as the previous sections. It looks like two huge paragraphs and many sentences are disconnected from the context. For example:
- Lines 178 to 180 and 159 to 161 state the aim of the study with redundancy.
- Lines 163 & 164: “..Thus, our study exclusively assessed the effect of cavity design ..”: Actually your results showed considerable differences between both types of clamps as well.
- Lines 225-228 state how different factors can affect the clamps and their behaviors. These lines should be added to the limitations of the study rather than the conclusion. Maybe use them with lines 166 and 167 to form a paragraph about the limitations of the study. The limitations need to be better explained anyway.
- Line 205: Start a new paragraph as you shift topics from PCD to clamps.

Experimental design

The effort put into the study is much appreciated, however, there is much room for improvement. Though tooth fractures because of the rubber dam clamps are not that frequent, they’re still upsetting and sometimes result in drastic changes to the treatment plan.

Concerning methodology:

Validation of the FEA models (the tooth and the clamps) was needed and should be added. It’s generally critical and in this study it’s even more important because the forces to be applied were deducted via FEA as well. Additionally, in your case it’s easily doable.
The forces exerted by the clamps can be registered via simple strain gauges placed on the original tooth or even another tooth with comparable geometry. If the values you deducted via FEA came out to be comparable to the ones registered on the actual tooth using the actual clamps (within an acceptable margin of error), it will add much value to your work.

I understand that vM is easier to understand and represent, but it has been shown that it doesn’t reflect the behavior of the tooth on its own, and its results don’t always translate to fracture probabilities. Please, consider adding the safety factor of the various models as a numerical representation that can be directly compared between models.

The demonstrative figure 2A shows the clamp to be placed mainly on dentin and 2C shows it to be almost equally divided between enamel and dentin. So the fact that your results showed that stresses build-up more in enamel, needs thorough investigation and explanation. May be stresses are distributed in dentin on a wider area as shown by Nawar et al (https://doi.org/10.1016/j.joen.2023.07.022)? You can check that easily by checking cross-sections. The same article also shows the importance of marginal ridges as an integral part of the enamel protective dome, which probably explains why your models having no marginal ridges came last in your work as well. So, maybe consider adding a paragraph about that as well

Validity of the findings

The abstract is well-written except for its conclusion which needs to be rephrased:
1. As I explained earlier in detail, I’d rather you don’t use strong statements like “The use of SS clamps in rubber dam isolation on teeth with significant tissue loss could contribute to tooth fractures” in the conclusions section. Maybe change it to something like: “SS Clamps cause higher stresses when compared to PEEK clamps”?
2. “Compared to other cavity designs, the GEC model showed the lowest vM values for the forces applied by the PEEK and SS clamps”. This shouldn’t be in your conclusion because it doesn’t match what the priority is in your aim. This sentence will give the reader the vibe that a specific access design should be favored based on how it affects the tooth’s behavior under the clamp, which would make the study useless because an endodontist would not sacrifice better visibility, accessibility, debridement to choose an access cavity that can withstand a specific clamp but rather he/she would manipulate their isolation armamentarium (including clamp choice) to fit the tooth’s condition and not the opposite.

Concerning the discussion:
How do you reconcile the fact that you found negligible stresses in all tissues other than enamel with your explanation in lines: 193-195: (The variable results in our study can be attributed to the amount of tissue removed from the pericervical dentin area where the clamp was placed, which was greater for these designs compared to others)?
I suggest that the esteemed authors recheck stresses in dentin in the pericervical region of the tooth. Maybe the clamp forces were mainly applied to enamel hence the negligible values of stresses in dentin? Maybe the dentinal stresses were distributed over a wide surface area in dentin. To feel the issue I’m addressing here please check how lines 197 to 205 highlight the importance of the pericervical dentin when in fact your results section show none of that!
Understandably, the clamp can cause a tooth to fracture in clinical practice. However, the clamps’ stresses are static and the tooth is subjected to them for a limited time, so I’d rather have the narrative of the discussion change to consider such incidence as a circumstantial failure rather than saying that it can predispose to future failures. In other words, to have a statement like line 216: “..could be a prognostic factor that increases the rate of tooth fractures” based on the study in hand is truly far-fetched.

Concerning the conclusion:
Please remove the following sentence: line 235: “The use of SS clamps in isolation could be an etiological factor in the fracture of mandibular molar teeth”. It’s a leap to state such a definitive statement depending on what we have here.

Reviewer 2 ·

Basic reporting

The following manuscript has been submitted for a review:"Evaluation of the forces applied by rubber dam clamps on mandibular first molar teeth with different endodontic access cavities through finite element analysis". The article looks interesting, english style is suffiiciently fluent but some improvements are need before publication.
Title: Too long, better: "Evaluation of the forces applied by rubber dam clamps on mandibular first molar teeth with different endodontic access cavities. A 3D FEA study."
Abstract: it is concise, linear.
Introduction: the topic is introduced clearly. The aim appear coherent, but the null hypothesis is missing. Please fill the gap and later in the Discussion you will confirm or reject it. Why did you choice the mathematical approach of FEA to investigate this mechanical behavior of the teeth. Please explain it and reference as suggested the use of FEA in Dentistry to reinforce your approach. FEA is widely accepted in the scientific community and the reader needs to understand it: You could also discuss the proper use of this "in silico" technique versus "in vitro" or "in vivo" ones.
At line 59, in fact, is missing the introduction of FEA in different fields of dentistry. You can also increase the quality of the paper by referencing other studies in endodontics or orthodontics. The reader must know the method you have choosen is elective.. The aim looks ok but fill the null hyphotesis, please.
Materials and Methods. They are described well. Of course, considering the parameters of the analysis listed in TAb 2, linear, homogeneous and isotropic materials,it will be necessary in the Discussion and conclusion explain the range of application of the considerations because of it will limit the final quality of the anaysis itself (lines 120-122).
Results: they are partially explained . Please rewrite better them in according to Figures.
Discussion: It is necessary to compare and to discuss the Results with the cited references in the text. But for example expressions like "bio"-mechanical ..." are not right. FEA doesn't match biological test in this investigation: Only tension, stresses and tensile movements are to be considered, You considered the mechanical behavior of the models investigated. Line 221: Reccomendations are not indicated. Results suggest to take in consideration some analyses aspects but it is not convenient to extend without data validation any clinical sentence. Please re-write this final part.
Conclusions: OK
Reference: to upload the references suggested.

Experimental design

no comment

Validity of the findings

no comment

---

## Round 0.2 · Minor Revisions

Dear Author,

The reviewer appreciates the revisions and suggests a few minor changes: add a brief explanation for using strain gauges in the methodology, tone down the conclusion regarding cavity design statements, add vM distribution perspectives to figure legends and ensure correct figure numbering. Please check other FEA references from different sources too to improve your text as requested by the first review round.

Kind regards,
Dr. Tribst JPM

Reviewer 1 ·

Basic reporting

The effort put into the revision is much appreciated, I hope the esteemed authors did not feel burdened by the requested modifications and I hope they feel their manuscript benefited from the revision. A few minor changes can be made:
Concerning the introduction:
The introduction benefitted a lot from the comments of the fellow reviewer.
Concerning methodology:
The validation: I’m aware that the advanced engineering calculations are not added to FEA dental papers. My request was a simple paragraph showing the simplified idea of how it's done like the one the authors added. So, if you don’t want to add a subsection about validation, I suggest adding a short sentence to the paragraph you added to explain why the strain gauges were used. For example, (To validate the forces estimates obtained via FEA, A strain gauge (BFLA-5-5, Tokyo Sokki Kenkyujo Co., Ltd., JP) was used in three directions to calculate the experimental stresses … etc), and please refer to Figure 4 in your text.

Experimental design

n/a

Validity of the findings

Concerning the conclusion
It still needs to be toned down a bit when it comes to statements about the cavity designs because as previously mentioned, the clamp placement is for an extremely limited duration and the forces are not occlusal by any means.

Additional comments

Concerning results:
Please add the perspective of the models showing the vM distribution to the figures' legends as well. The reader should be able to comprehend that from the figure legend without having to go back to the full text. Also, please make sure that the numbering of the figures matches where they refer in the manuscript. Either they’re mismatched or I’m having some technical issues with the revision files so please revise it again just in case.
Concerning the discussion:
Add the word “mostly” or any of it synonyms to line 222 to solve any remaining discrepancy between the points of load application in the figures: (As the application points of the clamps were MOSTLY on the enamel tissue of the tooth).
I recommend removing this sentence (lines 267 to 269) from the discussion as long as the differences are within the acceptable margin of error: (The differences between the FEA and strain gauge results are thought to be due to the limitation in obtaining point data on the tooth surface due to the fact that the strain gauge adhered to the tooth covers a large part of the tooth surface).
Correct the typo in line 278 (preservical).

---

## Round 0.3 · accepted · Accept

Dear Authors,

I am writing to inform you about the status of your manuscript titled "Evaluation of the forces applied by rubber dam clamps on mandibular first molar teeth with different endodontic access cavities. A 3D FEA study" submitted to PeerJ.

Upon review, I am pleased to confirm that all the reviewers' comments have been thoroughly addressed in your revised submission. Based on my evaluation, I am happy to report that the current version of your manuscript meets all necessary standards and criteria.

Therefore, I am pleased to inform you that your manuscript is ready for publication.

Congratulations on your work and thanks for choosing PeerJ.

Best regards,